# Rational design of pyrrole derivatives with aggregation-induced phosphorescence characteristics for time-resolved and two-photon luminescence imaging

Jianhui Yang[1,5], Yahui Zhang[2,5], Xinghui Wu[1], Wenbo Dai[1], Dan Chen[3], Jianbing Shi[1], Bin Tong[1], Qian Peng[4], Haiyan Xie[2], Zhengxu Cai [1✉], Yuping Dong [1] & Xin Zhang [3✉]

Pure organic room-temperature phosphorescent (RTP) materials have been suggested to be promising bioimaging materials due to their good biocompatibility and long emission lifetime. Herein, we report a class of RTP materials. These materials are developed through the simple introduction of an aromatic carbonyl to a tetraphenylpyrrole molecule and also exhibit aggregation-induced emission (AIE) properties. These molecules show non-emission in solution and purely phosphorescent emission in the aggregated state, which are desirable properties for biological imaging. Highly crystalline nanoparticles can be easily fabricated with a long emission lifetime (20 µs), which eliminate background fluorescence interference from cells and tissues. The prepared nanoparticles demonstrate two-photon absorption characteristics and can be excited by near infrared (NIR) light, making them promising materials for deep-tissue optical imaging. This integrated aggregation-induced phosphorescence (AIP) strategy diversifies the existing pool of bioimaging agents to inspire the development of bioprobes in the future.

---

[1] Beijing Key Laboratory of Construction Tailorable Advanced Functional Materials and Green Applications, School of Materials Science and Engineering, Beijing Institute of Technology, Beijing, China. [2] School of Life Science, Advanced Research Institute of Multidisciplinary Science, Beijing Institute of Technology, Beijing, China. [3] Department of Gynaecology, Cancer Hospital of China Medical University, Liaoning Cancer Hospital, Shenyang, People's Republic of China. [4] School of Chemical Sciences, University of Chinese Academy of Sciences, Beijing, People's Republic of China. [5] These authors contributed equally: Jianhui Yang, Yahui Zhang. ✉email: caizx@bit.edu.cn; zhangxiangmiao@hotmail.com

Organic fluorescence probes with aggregation-induced emission (AIE) characteristics are indispensable tools in biomedical science due to their nature to turn on their fluorescence, high photobleaching threshold, and large Stokes shift[1–5]. Their excellent biocompatibility also provides substantial opportunities for biological process monitoring and image-guided surgery[6,7]. However, interference from tissue autofluorescence, resulting in a low signal-to-noise ratio, limits the development of bioimaging with AIE materials[8,9]. Recently, pure organic room-temperature phosphorescent (RTP) materials have been described to be promising bioimaging materials due to their good bio-compatibility and long emission lifetime[10–13]. Afterglow imaging offers a stable luminescent signal that may eliminate interference from the autofluorescence of organisms. In addition, the triplet energy level of luminescent materials is lower than the singlet energy level, which endows these materials with a larger Stokes shift[12–15]. Therefore, combining AIE and RTP materials can take advantage of both the turn on nature and long lifetime properties. Although many RTP molecules have been developed in recent years, most of them are based on limited phosphor skeletons, such as carbazole and benzophenone[16–20]. These RTP molecules do not possess the potential for AIE. To overcome this problem and obtain RTP materials, the development of a versatile synthetic strategy based on the AIE skeleton is needed so that the development of molecules with aggregation-induced phosphorescence (AIP) properties can be explored. In general, there are three main prerequisites for AIP: functional groups favouring the n–π* transition[21–25], a proper molecular rotor that consumes the excited energy in solution[26–30], and special packing in the aggregated state for stabilization of the excited triplet state as well as the restriction of intramolecular motion[31–34]. Rationally optimized structures allow traditional AIE molecules to promote the intersystem crossing (ISC) process, resulting in phosphorescence emission instead of fluorescence emission. Our group has developed a series of aryl-substituted pyrrole derivatives and determined that aryl substitutions at different positions exhibit various properties[35]. Tetraphenylpyrrole (TePP, Fig. S1), without a phenyl ring at the N-position, exhibits intense fluorescence in both the solution and aggregated states. This is because the phenyl ring at the N-position acts as a rotor to consume the excited energy. Therefore, this introduction of a rotor can facilitate the n–π* transition and provide an opportunity to construct AIP materials for bioimaging.

Previous results have demonstrated that the incorporation of heavy atoms, heteroatoms, or aromatic carbonyls can boost spin–orbital coupling (SOC) and subsequently promote ISC processes[36–40]. In this work, we developed two RTP compounds,

phenyl-(2,3,4,5-tetraphenyl-1H-pyrrol-1-yl) methanone (TPM) and (4-chlorophenyl)-(2,3,4,5-tetraphenyl-1H-pyrrol-1-yl) methanone (TPM-Cl) (Fig. 1), on the basis of the TePP molecule. By introducing an aromatic carbonyl group at the N-position, spin–orbital coupling is initiated, and intersystem crossing (ISC) is enhanced. Since aromatic carbonyls can act as rotors to consume excited energy, the obtained compounds preserve the AIE character while simultaneously generating phosphorescent emission. Moreover, the presence of TPM-Cl can modulate the intermolecular packing, thus tuning phosphorescence performance[41–43]. Crystallization is an effective strategy to increase the luminescence efficiency of many AIEgens, as the dense packing of these molecules minimizes their intramolecular motion through crystallization[44,45]. The TPM/TPM-Cl prepared nanoparticles show phosphorescence quantum yields comparable to those of crystalline powders due to their high crystallinity. Time-dependent density functional theory (TD-DFT) calculations have proven that the introduction of aromatic carbonyls enhances ISC, which increases pyrrole derivative phosphorescence performance. Time-resolved imaging results have shown potential applications in the biological field. Furthermore, the prepared AIP nanoparticles show two-photon absorption (TPA) characteristics, which makes it possible to achieve RTP via two-photon near infrared (NIR) light excitation, as NIR-light excitation ensures lower phototoxicity and higher penetrability for cells and tissues[46,47].

## Results and discussion

**Synthesis and photophysical properties.** TPM and TPM-Cl were prepared by a one-step reaction from TePP in yields of 23% and 24%, respectively. The structures were confirmed by $^1$H and $^{13}$C NMR, mass spectrometry, and single-crystal X-ray diffraction (XRD). Their purities were determined by elemental analysis and high-performance liquid chromatography (HPLC). Figure 2 shows the UV–Vis absorption spectra and emission spectra of TPM and TPM-Cl. Compared to TePP and pentaphenylpyrrole (PPP), both TPM and TPM-Cl showed longer emission wavelengths. The Stokes shift of TPM is 232 nm, which is much larger than that of TePP or PPP (Fig. S2). Owing to their similar chemical structures, it is not plausible to attribute the significant Stokes shift to variations in the energy gap between the lowest singlet excited state ($S_1$) and ground state ($S_0$). We therefore propose that the yellow emission corresponds to the lowest triplet excited state ($T_1$), which is lower than $S_1$. To confirm this hypothesis, delayed spectra (delayed 50 μs) are shown in Fig. 2, where the curves are identical to their steady-state emission spectra, revealing the long lifetime characteristics of both TPM and TPM-Cl. The lifetimes of the TPM and TPM-Cl crystalline powders are 20.1 and 8.9 μs, respectively (Fig. S3 and Table S1). The introduction of a Cl atom into the phenyl ring only slightly increases the quantum yield but shortens the lifetime due to the heavy atom effect of the ISC process[48–50]. In addition, both the emission intensities and lifetimes of TPM/TPM-Cl increase with decreasing temperature from 290 to 80 K (Figs. S4 and S5). These results demonstrated that TPM and TPM-Cl exhibit pure phosphorescent emission.

To further verify the AIP properties of both TPM and TPM-Cl, emission spectra were recorded in water/THF mixtures with different water volume fractions ($f_w$) (Fig. 2b, e). When the water content is <60%, TPM/TPM-Cl is almost non-emissive, and no photoluminescence signal is observed. However, above 60% water content, the light intensity increases rapidly with increasing water content. This can be attributed to the formation of nanoaggregates in the poor solvent, restricting intramolecular motion. The aggregates showed their strongest emission intensity when the water content reached 80% with lifetimes of 18.5 μs (TPM) and

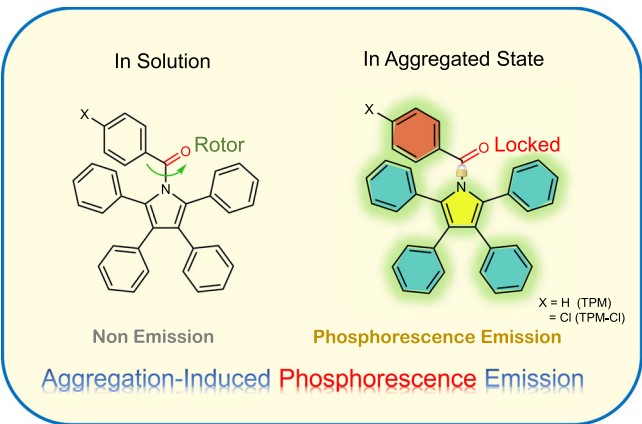

**Fig. 1 Molecular design strategy.** The molecular design strategy for the AIP compounds.

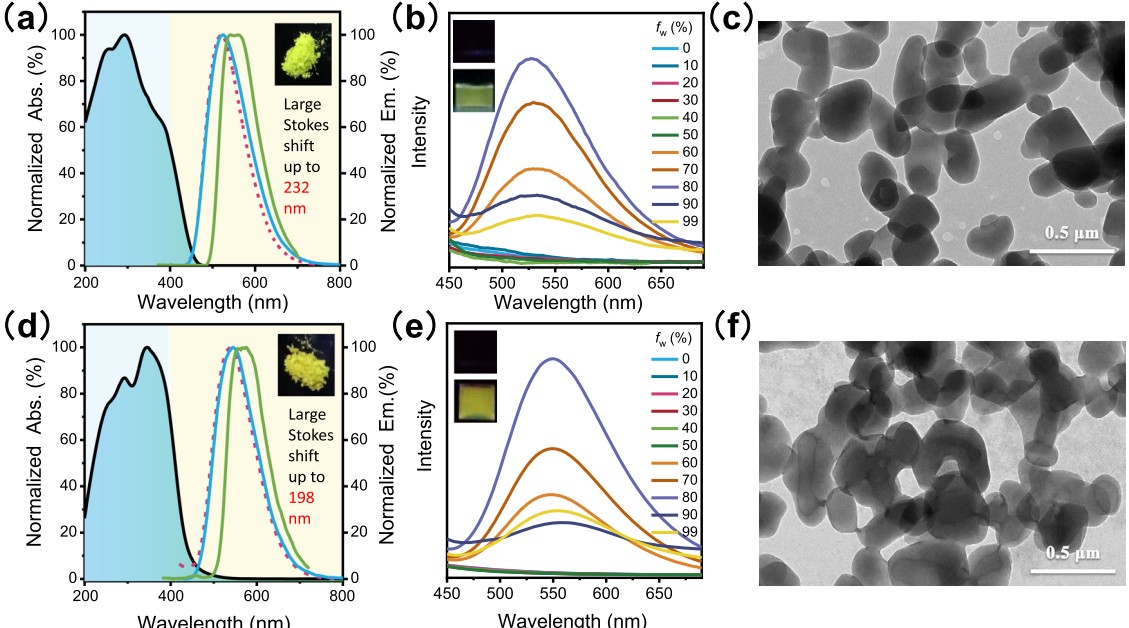

**Fig. 2 Photophysical properties and nanoparticle morphology.** UV absorption spectra in the solid-state (black line), steady-state emission spectra in the solid-state (red line) and phosphorescence spectra (50 μs delayed) in the solid-state (blue line) and THF at 77 K (10 μM, green line) of **a** TPM and **d** TPM-Cl. Abs.: absorption, Em.: emission. Inset: photographs of TPM/TPM-Cl as crystalline powders under 365 nm light irradiation. Emission spectra of **b** TPM and **e** TPM-Cl nanoparticles with different water fractions in H₂O/THF mixtures. $f_w$: water volume fraction. Inset: photographs of nanoparticles with 0% (up) and 80% (down) water fractions under 365 nm light irradiation. TEM images of the **c** TPM and **f** TPM-Cl nanoparticles. Each experiment was repeated three times independently.

7.4 μs (TPM-Cl), which were slightly shorter than those of the corresponding solid-states (Table S2). When the water fraction was greater than 80%, quick formation of the nanoaggregates led to amorphous packing, which reduced the stability of the triplet excitons (Fig. S10). Therefore, both the lifetimes and quantum yields of the nanoparticles with a high water content were lower than those at an 80% water content (Table S2 and Fig. S8).

Nanoparticle size was determined by transmission electron microscopy (TEM) (Fig. 2c, f). At 80% water content, uniform and stable nanorods formed. The nanoparticle sizes were in the submicrometre range (150–200 × 250–500 nm), which is suitable for biological imaging applications. Dynamic light scattering (DLS) showed that the nanoparticle sizes were 400 nm, which is consistent with the TEM results (Fig. S9). Generally, excited triplet energy can easily transfer to oxygen, subsequently quenching phosphorescence. A highly crystalline powder can act as an oxygen barrier to eliminate afterglow quenching. Therefore, phosphorescence performance strongly depends on the packing pattern. Previous results of investigated organic RTP nanoparticles showed that they were amorphous after being prepared by nanoprecipitation methods[51,52]. Compared to solid crystals, the phosphorescent properties of such amorphous nanoparticles can be greatly weakened. Interestingly, the XRD spectra showed that the TPM and TPM-Cl nanoparticles are highly crystalline, in which their fine crystal structure is comparable to that of the solid powders (Fig. S14). Good crystallinity provides intense intermolecular interactions and prevents phosphorescence quenching from environmental oxygen. Therefore, both the phosphorescence quantum yields and lifetimes of the TPM/TPM-Cl nanoparticles in water did not significantly decrease (Table S2). The ease of preparation, suitable size, and long lifetime of these TPM-based nanoparticles provide a great opportunity for zero-background time-resolved bioimaging.

**Single-crystal analysis and theoretical calculations.** Single crystals of both TPM and TPM-Cl were obtained via slow evaporation of a $CH_2Cl_2$/hexane solution. As shown in Fig. 3, the TPM and TPM-Cl molecules exhibited highly twisted conformations. The twist angle ($\theta$) between the pyrrole ring and phenyl group at the N-position of TPM is much larger (69.3°) than that of TPM-Cl (35.6°). Face-to-face molecular packing was not observed in the crystals of either TPM or TPM-Cl. This packing mode effectively avoids the ACQ effect caused by strong π–π interactions from large planar structures. The distances between the aromatic rings were all >3.6 Å (4.058–7.550 Å) and are relatively staggered, indicating the absence of π–π interactions. Additionally, the rotors in TPM and TPM-Cl (aromatic carbonyls) at the N-positions were examined, and it was found that the rotors were rigidly restricted by multiple C-H⋯π (2.848–2.987 Å) and C-H⋯O (2.435–2.755 Å) interactions around the molecules. These intermolecular interactions restrict rotor rotation and reduce non-radiative decay. In the case of TPM, one C-H⋯π interaction (2.848 Å) and three C-H⋯O interactions (2.590, 2.587, and 2.590 Å) were observed, while one C-H⋯π (2.987 Å) and only two C-H⋯O (2.435 and 2.755 Å) interactions were observed for TPM-Cl. Moreover, TPM processes two strong C-O⋯π interactions (3.079 Å) between the aromatic carbonyl at the N-position and two adjacent molecules. As a result, owing to the stronger restriction of the aromatic carbonyl at the N-position of TPM, the phosphorescence emission lifetime is considerably greater than that of TPM-Cl. As shown in Fig. S16, the shortest length that one TPM molecule occupies in the axial direction (a axis) is 6.591 Å, whereas this value for TPM-Cl is 12.256 Å (c axis). Hence, the molecules in the TPM crystal are more densely arranged than those in the TPM-Cl crystal, which promotes additional intense intermolecular interactions and suppresses the non-radiative transition of the excited triplet state. As a result, the measured phosphorescence lifetime of TPM is twice that of TPM-Cl.

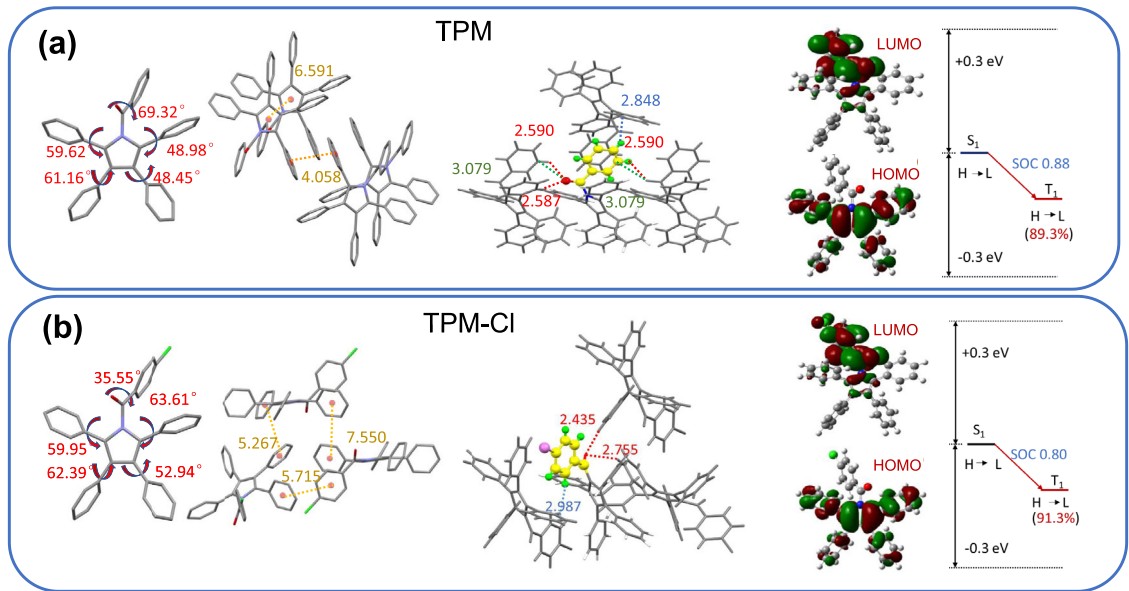

**Fig. 3 Single-crystal analysis and theoretical calculations.** Molecular conformations, intermolecular interactions, and molecular orbitals of the **a** TPM and **b** TPM-Cl crystals. Intermolecular interactions: aromatic ring interactions (orange lines), C-H⋯π (blue lines), C-H⋯O (red lines), and C-O⋯π (green lines). LUMO: lowest unoccupied molecular orbital, HOMO: highest occupied molecular orbital, SOC: spin–orbital coupling.

To determine the difference in phosphorescence performance between the polyarylpyrrole derivatives, DFT and TD-DFT calculations were performed using the Gaussian 09 program. The energies of the lowest unoccupied molecular orbital (LUMO) and highest occupied molecular orbital (HOMO) are presented in Fig. 3 and Fig. S17. The HOMOs of the polyarylpyrrole derivatives TePP, PPP, TPM, and TPM-Cl are similar and localized on the pyrrole and phenyl rings at the 2-, 3-, 4-, and 5-positions. However, the LUMOs are quite different; for TePP and PPP, localization occurs on the pyrrole and phenyl rings at the 2,5-positions or 1,2,5-positions and localization of TPM and TPM-Cl is mainly on the aromatic carbonyl at the N-positions. Therefore, the LUMOs and HOMOs of TPM and TPM-Cl are spatially separated, while those of TePP and PPP are partially separated. This high orbital separation can narrow the $\Delta E_{ST}$ ($\Delta E_{ST}$: energy gap between the lowest singlet and triplet states)[50,53]. The calculated $\Delta E_{ST}$ values for TPM and TPM-Cl (TPM, $\Delta E_{ST} = 0.13$ eV; TPM-Cl, $\Delta E_{ST} = 0.12$ eV) were considerably smaller than those of TePP and PPP (TePP, $\Delta E_{ST} = 1.11$ eV; PPP, $\Delta E_{ST} = 0.92$ eV). The smaller $\Delta E_{ST}$ values of TPM and TPM-Cl would largely promote the ISC rate between the singlet–triplet states, owing to $k_{ISC} \propto |SOC|^2 \exp\left[\frac{(\Delta E_{ST}-\lambda)^2}{\lambda}\right]$, which facilitates the generation of RTP. Considering that the smaller energy gap (<0.3 eV) between $S_1$ and $T_n$ can also be a possible ISC channel, the excitation energy and transition properties of all triplets with $\Delta E_{ST} > 0$ are given in Fig. 3, Fig. S17, and Tables S5–S8. The ISC processes of TPM and TPM-Cl are from $S_1$ to $T_1$ due to the narrow $\Delta E_{ST}$. However, no ISC channels were observed for TePP, and only one minor transition contribution from $S_1$ to $T_6$ was observed for PPP. The SOC value of PPP (0.32 cm$^{-1}$) was also much smaller than those of TPM (0.88 cm$^{-1}$) and TPM-Cl (0.80 cm$^{-1}$). Therefore, the introduction of aromatic carbonyls plays a crucial role in decreasing the $\Delta E_{ST}$ and increasing the SOC, thus generating RTP.

**Application studies in time-resolved bioimaging.** Since TPM-based nanoparticles displayed enhanced AIP properties in aqueous solution, they are satisfactory candidates for cell imaging. When HeLa cells were incubated with TPM, almost 91% of the cells survived after 24 h of co-staining at a concentration of 10 μM, indicating the low cytotoxicity of TPM (Fig. S18); additionally, the phagocytosis rate was as high as 63.3% according to UV analysis (Fig. S19). For confocal laser scanning microscopy (CLSM) imaging, the cells were stained with both TPM (10 μM) and commercial LysoTracker, which targets lysosomes with red emission. The AIP signal was apparently sourced from the high phagocytosis rate and the phosphorescence properties, and the intensity correlation quotient (ICQ) of the two signals was 0.781 (Fig. 4a–c), indicating that TPM was mainly entrapped in lysosomes.

Compared with one-photon imaging, two-photon luminescence imaging excited by an NIR-I pulsed laser displays greater performance in terms of deeper penetration, lower photodamage, and a higher signal-to-noise ratio. We were glad to find that TPM has two-photon imaging ability. The CLSM images of HeLa cells after incubation with TPM (10 μM) for only 10 min are shown in Fig. 4g–i. The photoluminescence signals of TPM from two-photon imaging irradiated with an 800 nm laser displayed an almost identical distribution compared to that of one-photon imaging (405 nm) (Fig. 4d). Upon excitation at 800 nm, these nanoparticles in aqueous solution emitted bright photoluminescence with similar band shapes and positions as the emissions excited by 405 nm. As shown in Fig. 4e, the photoluminescence intensity increased with increasing laser power. The correlation between the laser power and emission intensity demonstrated the two-photon excitation process (Fig. 4f). The two-photon imaging capability of TPM enables organic RTP under NIR-light excitation in aqueous solution, making TPM a potential candidate for imaging at the cellular level as well as in tissues and living bodies.

The background fluorescence of living organisms seriously interferes with the specificity and signal-to-noise ratio during bioimaging, but the long emission lifetime of phosphorescence has a unique advantage in eliminating background fluorescence. As shown in Fig. 5, high-quality long-lived signals were obtained in HeLa cells after incubation with TPM (10 μM). According to the images without time delay, it can be clearly observed that the signal is consistent with the two-photon absorption imaging results (Fig. 5a). After a 1 μs time delay, the phosphorescence

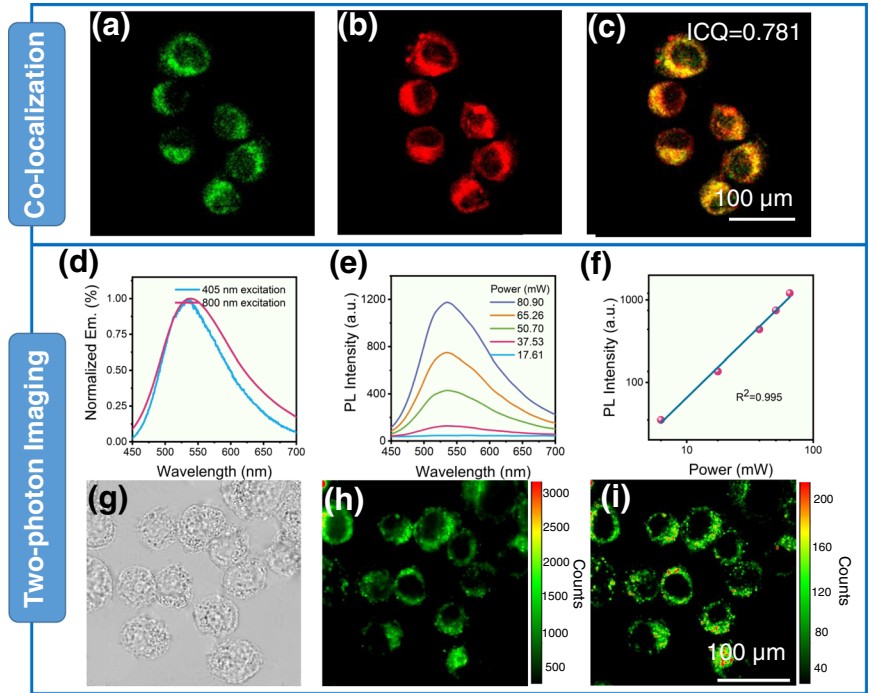

**Fig. 4 Co-localization and two-photon luminescence imaging.** CLSM images of **a** TPM, **b** LysoTracker, and **c** merged incubation with HeLa cells. ICQ: intensity correlation quotient. **d** Emission spectra of the TPM solid excited at 405 nm (blue line) and 800 nm (red line). **e** Emission spectra and **f** the logarithmic plot of the emission integral of the TPM solid at different excitation intensities (mW) initiated by an 800 nm femtosecond pulsed laser light. **g–i** CLSM images of HeLa cells incubated with TPM (10 μM in DMEM), Em.: Emission. **g** Bright field image; **h** 405 nm excitation, and **i** 800 nm excitation. Each experiment was repeated three times independently.

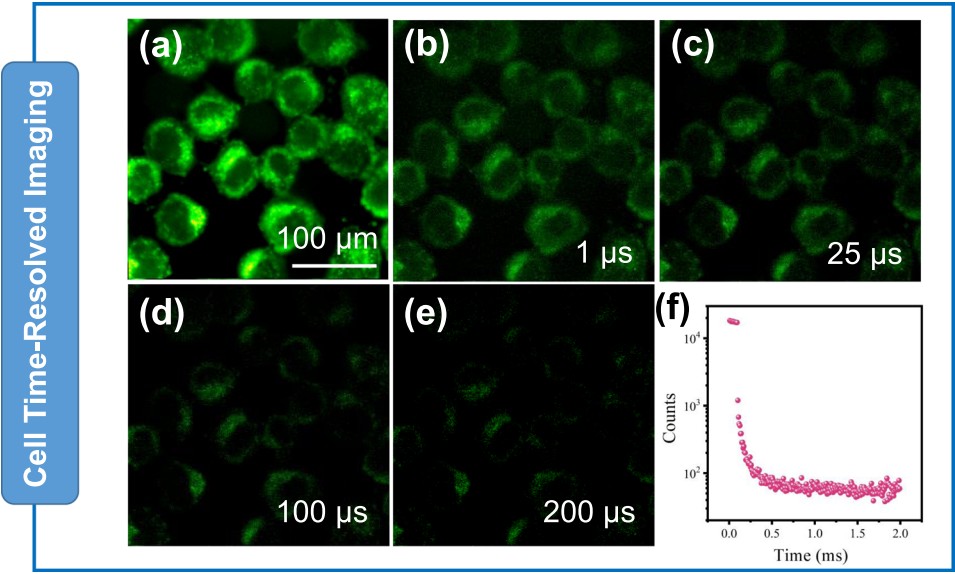

**Fig. 5 Time-resolved cellular imaging. a–e** Images of HeLa cells at different delay times. **f** Lifetime decay profile of the emission band in HeLa cells. HeLa cells were incubated with TPM (10 μM) at 37 °C for 10 min. The excitation wavelength was 405 nm. Each experiment was repeated three times independently.

signals of TPM are very clear, and even after 100 μs, the phosphorescence signals are still visible (Fig. 5d). The whole imaging process can last for more than 300 μs. As shown in Fig. 5f and Fig. S20, a high-quality long-lived signal was obtained in HeLa cells. Thus, TPM, which combines AIE and time-resolved luminescence, can effectively eliminate short-lived fluorescence interference by exerting a delay time, illustrating great potential in real imaging of complicated biological systems.

In vivo afterglow imaging was performed using an IVIS system in bioluminescence mode. As shown in Fig. 6a, a phosphorescent signal from TPM nanoparticles could still be detected at 6 min after the removal of light excitation. On the basis of this phenomenon, afterglow imaging in anaesthetized living nude mice was further investigated. TPM (47.5 μg/mL, 100 μL) was subcutaneously injected into nude mice. After 60 s, the handheld UV lamp was turned off (365 nm), and the in vivo

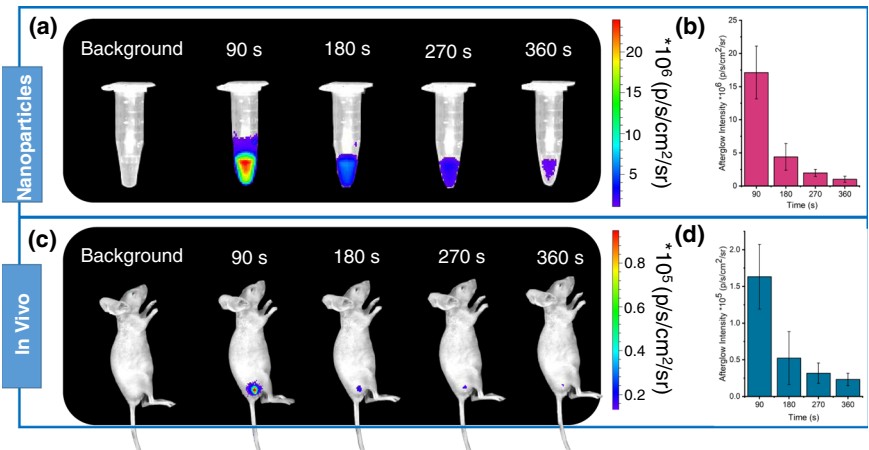

**Fig. 6 In vivo afterglow imaging. a** Afterglow images after UV irradiation (25.5 mW/cm$^2$) of TPM (47.5 μg/mL) for 60 s. **b** Quantification of the afterglow intensities of the sample in **a**. **c** In vivo afterglow imaging of living mice after subcutaneous injection (100 μL) of TPM (47.5 μg/mL). Afterglow images were acquired after UV irradiation of mice for 60 s. **d** Quantification of the afterglow intensities of the injection areas in **c**. All bars represent mean ± SD (n = 3).

phosphorescence imaging results revealed that the phosphorescence signals from the TPM nanoparticles could be observed at 6 min after the removal of light excitation (Fig. 6c). The signal-to-background ratios (SBRs) reached 7.14 ± 0.65, indicating the advantage of time-resolved phosphorescence imaging with negligible background interference.

In summary, we developed a class of AIP molecules by introducing an aromatic carbonyl group to the TePP molecule. Spectroscopic analysis and single-crystal structure analysis demonstrated that RTP originates from the aggregation of molecules. Bright nanoparticles were prepared by a simple nanoprecipitation method and used in the time-resolved bioimaging of HeLa cells and living nude mice with a high signal-to-noise ratio. Two-photon absorption characteristics made excitation by NIR-light accompanied by lower phototoxicity and higher penetrability possible. The long lifetime, low cytotoxicity, and NIR irradiation properties of TPM allow the utilization of pure organic phosphors for the development of biological theranostics, including, but not limited to, imaging-guided surgery, endoscopic examination, and in vivo imaging.

## Methods

**Synthesis of TPM and TPM-Cl**. A flask was charged with 2,3,4,5-tetraphenyl-1H-pyrrole (TePP) (1.5 mmol), benzoyl chloride derivatives (1.5 mmol), triethylamine (1.5 mmol), 4-dimethylaminopyridine (3.5 mmol) and dichloromethane (3 mL). The flask was then briefly evacuated and backfilled with nitrogen for three times. The mixture was stirred for 12 h at 40 °C under nitrogen atmosphere. After removal of solvent under reduced pressure, the crude products were purified by column chromatography (petroleum ether/dichloromethane 3:1, v/v) to afford product.

TPM, yellow solid, 23% yield. $^1$H NMR (CDCl$_3$, 400 MHz). δ: 7.69–7.71 (d, 2H), 7.34–7.37 (t, 1H), 7.19–7.23 (t, 2H), 7.08–7.12 (m, 16H), 6.96–6.99 (m,4H). $^{13}$C NMR (CDCl$_3$, 100 MHz). δ: 171.05, 134.92, 134.27, 133.46, 132.10, 131.98, 130.67, 130.46, 128.18, 127.94, 127.69, 127.12, 126.01, 124.66. HRMS (m/z): calcd. for C$_{35}$H$_{25}$NO 476.20089, found 476.19995. Anal. calcd. for C$_{35}$H$_{25}$NO: C 88.39, H 5.30, N 2.95; found: C 88.12, H 5.06, N 3.09.

TPM-Cl, yellow solid, 24% yield. $^1$H NMR (CDCl$_3$, 400 MHz). δ: 7.62–7.64 (d, 2H), 7.17–7.19 (d, 2H), 7.09–7.12 (m, 16H), 6.94–6.97 (m, 4H). $^{13}$C NMR (CDCl$_3$, 100 MHz). δ: 169.98, 139.91, 134.06, 133.31, 132.01, 131.92, 131.79, 130.92, 130.39, 128.55, 128.05, 127.71, 127.30, 126.10, 124.90. HRMS (m/z): calcd. for C$_{35}$H$_{24}$ClNO 510.16192, found 510.16125. Anal. calcd. for C$_{35}$H$_{24}$ClNO: C 82.42, H 4.74, N 2.75; found: C 82.26, H 4.56, N 2.52.

**Crystal growth**. All single-crystal samples were obtained from slow evaporative crystallization using a hexane/dichloromethane mixture (1:1, v/v).

**Animals**. Six-weeks-old BALB/c nude mice were purchased from the Vital River Laboratory Animal Technology Co., Ltd. (Beijing, China).

*Statement of ethical approval*. All animal studies were performed in accordance with the Regulations for Care and Use of Laboratory Animals and Guideline for Ethical Review of Animals (China, GB/T 35892-2018) and the overall project protocols were approved by the Animal Ethics Committee of Beijing Institute of Technology. The accreditation number is BIT-EC-SCXK(Jing) 2019-0010-M-2020019 promulgated by Animal Ethics Committee of Beijing Institute of Technology.

*Feeding conditions*. All the animals were submitted to controlled temperature conditions (22–26 °C), humidity (50–60%) and light (12 h light/12 h dark, 15–20 LX). They had access to water and food ad libitum in barrier system of Beijing Institute of Technology. (SYXK (Jing) 20170013).

**Reporting summary**. Further information on research design is available in the Nature Research Reporting Summary linked to this article.

## Data availability

The authors state that the data supporting the results of this study are available in this paper and its supplementary materials. Extra data are available from the corresponding authors upon reasonable request. The data generated in this study have been deposited in the Figshare database: https://doi.org/10.6084/m9.figshare.14913162.v1. The X-ray crystallographic coordinates for structures reported in this study have been deposited at the Cambridge Crystallographic Data Centre (CCDC), under deposition numbers 1975269 and 1975270. These data can be obtained free of charge from The Cambridge Crystallographic Data Centre via www.ccdc.cam.ac.uk/data_request/cif.

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

## Acknowledgements

This work was financially supported by the National Natural Scientific Foundation of China (Grant Number: 21975021, 51803009, 21905021, 21975020, and 21875019). This work was also supported by Beijing National Laboratory for Molecular Sciences (BNLMS202007), China Postdoctoral Science Foundation (2019TQ0034) and Liaoning Key Research & Development Program (2019JH8/10300073).

## Author contributions

J.Y. synthesized all materials, Y.Z. and J.Y. performed photophysical measurements, conducted the bioimaging experiments, and prepared the paper. Thus, J.Y. and Y.Z. contributed equally to this work. X.W. and Q.P. performed the theoretical calculations. W.D. and D.C. helped the measurements of lifetime. J.S., B.T. and H.X. helped the characterizations of bioimaging and structure. Z.C., Y.D. and X.Z. designed and supervised the research and wrote the paper. All authors commented on the paper.

## Competing interests

The authors declare no competing interests.
