## [Peer Review File · Nature Communications]

Reviewers' comments:

Reviewer #1 (Remarks to the Author):

In this manuscript, Cai and coauthors reported a series of pyrrole derivatives with phosphorescence by introduction of aromatic carbonyls in solid state. The photophysical properties of nanoparticles in solution were also investigated. Moreover, the potential applications for time-resolved and two-photon luminescence imaging were demonstrated. Regarding the previous work (J. Am. Chem. Soc. 2019, 141, 5045–5050), this manuscript is not innovative enough to the high-level journal of Nature Communications. So I do not recommend its publication in Nature Communications.

1. The authors state "When water fraction was higher than 80%, quick formation of the nanoaggregates leads to amorphous packing". It is needed to provide the evidence of the phase transition, such as XRD or electron diffraction patterns.
2. It is well known that the photoluminescence is highly dependent morphology of materials. How about the photoluminescence properties in PMMA and amorphous states?
3. From the TEM images, the nano-aggregates are not well-dispersed, which will greatly influence the biocompatibility and imaging quality.
4. There is no animal experiments. For a good bio-probe, the imaging in vivo is needed.

Reviewer #2 (Remarks to the Author):

This manuscript reports two new organic molecules based on TEPP with room temperature phosphorescence (RTP) property. Bright nanoparticles were prepared and used in time-resolved bioimaging of HeLa cells by two-photon excitation with NIR light. The topic of this manuscript is very interesting and significant. It can be published after some minor revisions.

1. The AIE property of Phenothiazine has been proven in many papers, such as Adv. Mater. 2020, 32, 1903530. So, it is wrong that this work claims phenothiazine doesn't possess potential AIE property.
2. The strategy to realize AIE and RTP simultaneously in this paper is interesting and novel. More molecules may be synthesized to testify the universality of this strategy.
3. The authors emphasized the carbonyl benzene at N-position of molecules acts as a rotor to consume the excited energy in solution state. That is to say, the AIE property of target molecules is derived from the restriction of motions of the carbonyl benzene at N-position. However, the target molecules also have four phenyl rings can be rotated freely. Except for the reference cited, more computational and experimental study should be carried out to discuss the importance of the carbonyl benzene.
4. It is not suitable to discuss between the absorption spectra in solution state and the emission spectra in solid states in Figure 1. The absorption spectra of solid states should be used.
5. The emission spectrum and lifetime of amorphous states are required. If the RTP is disappeared in the amorphous state, crystallization induced phosphorescence (CIP) is a more accurate expression than aggregation induced phosphorescence (AIP) to describe the phenomenon of this work.
6. The original temperature-dependent lifetime spectra of TPM/TPM-Cl are required.
7. SAED pattern of nanocrystals from TEM (Figure 1) are required.
8. The preparation method of nanocrystals used for PXRD (Figure S7) are required.
9. The details of HPLC measurement are required, such as the detection wavelength.
10. The stability of nanocrystals is required.
11. The authors can describe more detail to demonstrate the two-photon excitation process (Figure 3E and Figure 3F).
12. The authors claim that TPM can be mainly entrapped into lysosome in RAW 264.7 cells. Can TPM be entrapped into lysosome in HeLa cells? Please show some experimental data.
13. Images of HeLa cells at different delay time irradiated with 800 nm laser are required.

Reviewer #3 (Remarks to the Author):

Manuscript "Rational Design of Pyrrole Derivatives with Aggregation-Induced Phosphorescence Characteristic for Time-Resolved and Two-Photon Luminescence Imaging" by Xin Zhang et al. reports on tetraphenylpyrrole derivatives displaying aggregation-induced emission and room-temperature phosphorescence which is maintained in highly crystalline nanoparticles. In addition, the nanoparticles show two-photon absorption properties.

The work is well performed and, except for some sentences (i.e. ...and all do not possess potential AIE property..., ... The Stokes shift of TPM..., ... such amorphous nanoparticles can be greatly reduced or even disappeared... just to mention few examples), well and clearly written.

The subject is certainly of large interest; however I have some concerns regarding its novelty.

The main interesting aspect of the work is certainly to be associated with the high crystallinity of nanoparticles (allowing their use in bioimaging) which are usually obtained as "Previous results of organic RTP nanoparticles were amorphous once prepared by the nanoprecipitation

methods.^{49,50}". If this study represents a true novelty on this point, this should be emphasized by indicating that this is the first example or one of the few examples (citing the proper literature).

The sentence "To our delight, the prepared nanoparticles of TPM and TPM-Cl show comparable phosphorescence quantum yield as crystalline powders due to high crystallinity." should be supported with proper references. As mentioned, I would substitute "with our delight" with proper examples...

Response to Reviewer 1

In this manuscript, Cai and coauthors reported a series of pyrrole derivatives with phosphorescence by introduction of aromatic carbonyls in solid state. The photophysical properties of nanoparticles in solution were also investigated. Moreover, the potential applications for time-resolved and two-photon luminescence imaging were demonstrated. Regarding the previous work (J. Am. Chem. Soc. 2019, 141, 5045–5050), this manuscript is not innovative enough to the high-level journal of Nature Communications. So I do not recommend its publication in Nature Communications.

Response: We thank the reviewer's suggestion.

Actually, the novelty and significance of our work were different from the previous work. First, although some room temperature phosphorescence (RTP) molecules have been developed in recent years, most of them were based on the limited skeletons, including carbazole, benzophenone, and phenothiazine (Angew. Chem. Int. Ed. 2019, 58, 17220; Angew. Chem. Int. Ed. 2018, 57, 8425; Nat. Commun. 2018, 9, 840; Angew. Chem. Int. Ed. 2019, 58, 6645). For example, the molecules in the literature as mentioned by the reviewer (J. Am. Chem. Soc. 2019, 141, 5045) were also developed by the combination of carbazole and difluoroboron β -diketonate. Actually, in addition to the carbazole moiety, difluoroboron β -diketonate was also a RTP skeleton. Oxygen-sensitive phosphorescence properties of difluoroboron β -diketonate based compounds were first reported by Prof. Cassandra L. Fraser (J. Am. Chem. Soc. 2007, 129, 8942) and used for tumor hypoxia imaging (Nat. Mater. 2009, 8, 747). Therefore, in the JACS work, they focused on the assembly to form nanoparticles (NPs) with strong intermolecular interaction, and perform RTP property. In our work, we developed a new class of RTP molecules based on pyrrole molecules. The RTP performance of pyrrole-based molecules has rarely been reported in previous results.

Secondly, in our work, we developed a new luminescence system, aggregation induced phosphorescence. The molecules showed non-emission in solution, but pure phosphorescence emission in aggregated state with a large Stokes shift. They can take both the advantage of the RTP (long life-time, zero background imaging) and AIE (turn-

on response, long retention time in living samples). The molecules were totally new and the properties have not been reported by previous work.

In addition, the nanoparticles have the ability of two-photon luminescence imaging, which enable organic RTP under NIR-light excitation (800 nm) in aqueous solution. Even though the JACS's work also carried out two photon luminescence imaging. Our materials are successfully applied for time-resolved bioimaging of HeLa cells and living nude mice with a high signal-to-noise ratio. The delayed time up to 360 s can effectively avoid the autofluorescence interference. However, time-resolved bioimaging was not achieved by the JACS work. Due to the high requirements of time-resolved imaging on RTP materials, only limited research groups with several works can achieve time-resolved imaging in vivo or in vitro (Adv. Mater. 2020, 32, 2006752; Angew. Chem. Int. Ed. 2020, 59, 9946; Adv. Mater. 2019, 31, 1807222; DOI: 10.1002/adma.202007811; Nat. Commun. 2018, 9, 840; Adv. Mater. 2017, 29, 1606665; Nat. Commun. 2020, 11, 842; ACS Appl. Mater. Interfaces 2020, 12, 18385). Therefore, the development of RTP materials for time-resolved imaging was still challenging.

1. The authors state “When water fraction was higher than 80%, quick formation of the nanoaggregates leads to amorphous packing”. It is needed to provide the evidence of the phase transition, such as XRD or electron diffraction patterns.

Response: We thank the reviewer's suggestion. Actually, this phenomenon was common in aggregation induced emission materials (J. Am. Chem. Soc. 2014, 136, 7383; Angew. Chem. Int. Ed. 2011, 50, 11654). The conclusion that amorphous packing led to low luminescence efficiency was also drawn by the previous results (Chem. Commun. 2007, 40; Chem. Commun. 2010, 46, 686). We also carried out the electron diffraction to further provide the evidence of the phase transition. The SAED (selected area electron diffraction) of TPM/TPM-Cl nanoparticles with different water fractions were shown in Figure S9. The diffraction spots can be observed in the nanoparticles that obtained from 80% water content. In contrast, only ring-like diffraction pattern can be observed in the nanoparticle that obtained from the 90% water content, indicating the lower crystallinity of nanoaggregates from higher water content solvents.

Figure S9. The SAED of TPM/TPM-Cl nanoparticles with 80% (A and C) and 90% (B and D) water fractions, respectively.

2. It is well known that the photoluminescence is highly dependent morphology of materials. How about the photoluminescence properties in PMMA and amorphous states?

Response: We thank the reviewer's suggestion. The photoluminescence properties of TPM/TPM-Cl in PMMA were carried out (Figure S14). No fluorescence or phosphorescence emission can be detected when both TPM and TPM-Cl were doped in PMMA. Generally, when AIE materials were added into a high viscosity solvent, such as glycerin, PMMA, the fluorescence of AIE materials will increase due to the restriction of the molecular motion. However, in our work, both the fluorescence and phosphorescence can not be observed. This is because our TPM and TPM-Cl were aggregation induced phosphorescence (AIP) materials. The generation of phosphorescence not only need the restriction of molecular motion, but also the formation of the nano-aggregates. Therefore, no fluorescence and phosphorescence of TPM and TPM-Cl can be observed in PMMA. This could be another evidence that both TPM and TPM-Cl showed AIP performance. In addition, the amorphous powders of TPM and TPM-Cl were difficult to be prepared due to the high crystallinity of TPM and TPM-Cl. XRD pattern of the grinding powder still showed the similar pattern as the crystalline powder (Figure S13). However, on the basis of question 1, we found the nanoparticles that obtained from higher water content showed a low crystallinity, resulting in low emission efficiency. Therefore, the photoluminescence properties of our materials were also highly dependent on the morphology of materials.

Figure S14. The fluorescence spectra and phosphorescence spectra of TPM and TPM-Cl in PMMA (PMMA $M_w = 120,000$, TPM/TPM-Cl: PMMA = 1: 100, mass ratio).

Figure S13. XRD patterns of TPM and TPM-Cl in crystalline powders, grinding powders, and nanoparticles.

3. From the TEM images, the nano-aggregates are not well-dispersed, which will greatly influence the biocompatibility and imaging quality.

Response: We thank the reviewer for pointing this out. The morphology of the nano-aggregates was highly dependent on the preparation methods. In our work, the sample were prepared in a mixture of THF/water. Both TPM and TPM-Cl showed nanoaggregates with a size around 400 nm in THF/water solution (Figure S8). However, when the mixture solution was transferred to the grids for TEM measurements, the little

amount of THF were evaporated quickly, to a certain extent, forming some poorly dispersed nanoaggregates. However, THF was not allowable for the bioimaging measurement. Therefore, the nanoaggregates were prepared in a mixture of dimethyl sulfoxide (DMSO) and phosphate buffered saline (PBS). The TEM images of nanoaggregates prepared in DMSO/PBS solution were also carried out (Figure S11). The nanoparticles were well-dispersed with a uniform size around 250 nm. Intense phosphorescence was also observed when the nanoparticles were formed (Figure S12). Since the nanoaggregates were prepared in DMSO/PBS solution for bioimaging, we believe the nanoparticle should not influence the biocompatibility and imaging quality. Generally, in the work about AIE, researchers preferred to report their AIE titration results in THF/water solution (Angew. Chem. Int. Ed. 2021, 60, 5386; ACS Nano 2018, 12, 7936; Adv. Mater. 2017, 29, 1604100; Angew. Chem. Int. Ed. 2015, 54, 7275; Adv. Funct. Mater. 2020, 30, 2002546). To make a better comparison, we also reported these results in THF/water solution.

Figure S11. TEM images of (A) TPM and (B) TPM-Cl nanoparticles in the DMSO/PBS mixtures (10 μM, DMSO: PBS = 10: 90, volume ratio).

Figure S12. Emission spectra of TPM/TPM-Cl nanoparticles with different water fractions in the DMSO/PBS mixtures.

4. There is no animal experiments. For a good bio-probe, the imaging in vivo is needed.
 Response: We thank the reviewer's suggestion. In vivo afterglow imaging of living mice after subcutaneous injection (100 μL) of TPM ($47.5 \mu\text{g mL}^{-1}$) was taken and shown in Figure 5.

We added:

Figure 5. (A) Afterglow images after UV irradiation (25.5 mW/cm^2) of TPM ($47.5 \mu\text{g/mL}$) for 60 s. (B) Quantification of afterglow intensities of sample in (A). (C) In vivo afterglow imaging of living mice after subcutaneous injection (100 μL) of TPM ($47.5 \mu\text{g/mL}$). Afterglow images were acquired after UV irradiation of mice for 60 s. (D) Quantification of afterglow intensities of injection areas in (C).

In vivo afterglow imaging was taken by IVIS living imaging system under bioluminescence mode. As shown in Figure 5A, phosphorescent signal from TPM nanoparticles could still be detected even at 6 min after the removal of light excitation. On the basis of this phenomenon, afterglow imaging in anesthetized living nude mice was further investigated. TPM (47.5 $\mu\text{g/mL}$, 100 μL) were subcutaneously injected to nude mice. At 60 s upon turning off handheld UV lamp (365 nm), in vivo phosphorescence imaging results revealed that phosphorescent signals from TPM nanoparticles can be observed at 6 min after the removal of light excitation (Figure 5C). The signal-to-background ratios (SBRs) can be reached up to 7.14 ± 0.65 , indicating the advantage of time-resolved phosphorescence imaging with negligible background interference.

Response to Reviewer 2

1. The AIE property of Phenothiazine has been proven in many papers, such as Adv. Mater. 2020, 32, 1903530. So, it is wrong that this work claims phenothiazine doesn't possess potential AIE property.

Response: We are sorry for the mistake. We have corrected it in the revised manuscript.

2. The strategy to realize AIE and RTP simultaneously in this paper is interesting and novel. More molecules may be synthesized to testify the universality of this strategy.

Response: We thank the reviewer's suggestion. To testify the universality of this strategy, two other molecules were synthesized (Scheme R1). The phosphorescence spectra (50 μs delayed) are shown in Figure R1, where the curves are identical to their steady-state emission spectra, revealing the RTP character of both TPM-1 and TPM-2. TPM-1/TPM-2 was almost non-emissive in THF solution. However, the aggregates showed the intense yellow emission when the water content reached 80%. All these data demonstrated the introduction of aromatic carbonyl to a tetraphenylpyrrole was an effective approach for the preparation of AIP molecules. More AIP cores were also currently pursued in our lab and will be reported at due time.

Scheme R1. Chemical structures of TPM-1 and TPM-2 molecules.

Figure R1. The steady-state spectra and phosphorescence spectra (50 μ s delayed) in solid states of TPM-1 (A) and TPM-2 (B). Inset: photographs of nanoparticles with 0% and 80% water fractions respectively under 365 nm light irradiation in the H₂O/THF mixtures.

3. The authors emphasized the carbonyl benzene at N-position of molecules acts as a rotor to consume the excited energy in solution state. That is to say, the AIE property of target molecules is derived from the restriction of motions of the carbonyl benzene at N-position. However, the target molecules also have four phenyl rings can be rotated freely. Except for the reference cited, more computational and experimental study should be carried out to discuss the importance of the carbonyl benzene.

Response: We thank the reviewer's suggestion. To explain the particularity of the substituents at N-position relative to the four phenyl rings at other positions, LUMO and HOMO energies of eight polyarylpyrroles are compared (Figure R2). All the LUMO energy levels are mainly localized at the phenyl ring at the 1-position, except

TePP and DPP. Therefore, when exciton was excited from the ground state to the excited state, it was difficult to relax back to the ground state by the radiative transitions in the single molecular state due to partial spatial separation. The phenyl group at the 1-position can be a rotor and the dynamic rotation non-radiatively dissipated the exciton energy. Therefore, all the molecules with the phenyl at the 1-position (TPM, TPM-Cl, PPP and TPP) showed AIE character. While molecules without phenyl groups at N-position, such as TePP, showed strong fluorescence in both solution and aggregated states ($\Phi_{PL} = 65.6\%$ in THF and $\Phi_{PL} = 74.3\%$ in solid state). Both computational and experimental results demonstrated the phenyl at N-position of molecules acts as a rotor to consume the excited energy in solution state. Therefore, TPM and TPM-Cl with carbonyl benzene at N-position were also non-emissive in dilute solution.

Figure R2. HOMO (left) and LUMO (right) energy levels of polyarylpyrroles.

4. It is not suitable to discuss between the absorption spectra in solution state and the emission spectra in solid states in Figure 1. The absorption spectra of solid states should be used.

Response: We thank the reviewer's suggestion. The absorption spectra of solid states were added in the revised manuscript.

Figure 1. UV absorption spectra in solid states (black line), steady state spectra in solid states (red line), phosphorescence spectra (50 μs delayed) in solid states (blue line) and in THF at 77 K (10 μM, green line) of (A) TPM and (D) TPM-Cl. Inset: photographs of TPM/TPM-Cl in crystalline powder under 365 nm light irradiation. Emission spectra of (B) TPM and (E) TPM-Cl nanoparticles with different water fractions in the H₂O/THF mixtures. Inset: photographs of nanoparticles with 0% (up) and 80% (down) water fractions respectively under 365 nm light irradiation. TEM images of (C) TPM and (F) TPM-Cl nanoparticles.

Figure S1. UV absorption spectra and fluorescence spectra in solid states at room temperature of TePP and PPP.

5. The emission spectrum and lifetime of amorphous states are required. If the RTP is disappeared in the amorphous state, crystallization induced phosphorescence (CIP) is a more accurate expression than aggregation induced phosphorescence (AIP) to describe the phenomenon of this work.

Response: We thank the reviewer's suggestion. The pure amorphous powders of TPM and TPM-Cl were different to be prepared due to the high crystallinity of TPM and TPM-Cl. XRD pattern of the grinding powder still showed the similar pattern as the crystalline powder (Figure S13). However, on the basis of question 1 (reviewer 1), we found the nanoparticles that obtained from higher water content, showed a low crystallinity, resulting in low emission efficiency. Therefore, we agreed that the phosphorescence performance should be determined by the crystallinity of the nanoparticles. However, since our materials showed phosphorescence performance when the formation of the aggregates, we would like to use the aggregation induced phosphorescence due to the ease formation of nanoaggregates with the phosphorescent property.

Figure S13. XRD patterns of TPM and TPM-Cl in crystalline powders, after grinding and nanoparticles.

6. The original temperature-dependent lifetime spectra of TPM/TPM-Cl are required.

Response: We thank the reviewer's suggestion. The original temperature-dependent lifetime spectra of TPM/TPM-Cl were added in Figure S5 and Figure S6 in revised manuscript.

We added:

Figure S5. The phosphorescence lifetime of TPM at different temperatures.

Figure S6. The phosphorescence lifetime of TPM-CI at different temperatures.

7. SAED pattern of nanocrystals from TEM (Figure 1) are required.

Response: We thank the reviewer's suggestion. SAED pattern of nanocrystals from TEM were added.

Figure S9. The SAED of TPM/TPM-CI nanoparticles with 80% (A and C) and 90% (B and D) water fractions, respectively.

8. The preparation method of nanocrystals used for PXRD (Figure S7) are required.

Response: We thank the reviewer's suggestion.

We added: The preparation method of nanocrystals used for PXRD: Nanocrystals used for PXRD were prepared by nanoprecipitation method in solvent/antisolvent medium. In detail, a stock solution of TPM/TPM-CI in THF (10^{-3} M, 1 mL) was injected rapidly into 10 mL water and vigorously stirred at room temperature for 10 min. Then the nanoparticles of TPM/TPM-CI were dried by freeze drying method for PXRD measurements.

9. The details of HPLC measurement are required, such as the detection wavelength.

Response: We thank the reviewer's suggestion. The detailed HPLC measurement were added in revised SI.

We added: Column: Accucore Vanquish C18+ 100 x 2.1 mm. Particle size:1.5 μm .

Mobile phase: acetonitrile. Flow rate: 0.2 mL/min. Column oven 40°C.

10. The stability of nanocrystals is required.

Response: We thank the reviewer's suggestion. The stability of nanocrystals was added in Figure S10. Nanoparticles show good stability. No obvious changes were observed in seven days.

Figure S10. The variation of particle sizes measured by dynamic light scattering (DLS) after storage for 1-7 days at room temperature.

11. The authors can describe more detail to demonstrate the two-photon excitation process (Figure 3E and Figure 3F).

Response: We thank the reviewer's suggestion. The detailed method was shown in SI "Two-photon imaging: HeLa cells were seeded in $\Phi 20$ mm glass bottom cell culture dishes ($1.0 \pm 0.05 \times 10^6$ cells in each dish). After overnight culture in a humidified incubator at 37 °C with 5% CO₂, culture medium was removed and cells were stained with 10 μM of TPM for 10 min in DMSO/DMEM (10% DMSO) solution at 37 °C. After washed by PBS for 3 times, HeLa cells were fixed with 4% fixative solution for 10 min. Before imaging, each dish was washed by PBS for 3 times. The two-photon imaging of TPM in cells was recorded with a Nikon scanning confocal microscope."

In addition, more details of two-photo excitation process were added in the revised

manuscript.

We added: Upon excitation at 800 nm, these nanoparticles in aqueous solution emit brightly with similar band shape and position as the emissions excited by 405 nm. As shown in Figure 3E, the photoluminescence intensity increases with the increasing of laser power. The dependence between the laser powers and emission intensities demonstrated the two-photon excitation process (Figure 3F).

12. The authors claims that TPM can be mainly entrapped into lysosome in RAW 264.7 cells. Can TPM be entrapped into lysosome in HeLa cells? Please show some experimental data.

Response: We thank the reviewer's suggestion. The experimental data of in HeLa cells was carried out and the images were updated in Figure 3A, B and C. As shown in Figure 3, TPM were also entrapped into lysosome in HeLa cells.

Figure 3. CLSM images of (A) TPM, (B) LysoTracker, (C) merge incubated with HeLa cells. (D) Emission spectra of TPM solid excited at 405 nm (blue line) and 800 nm (red line). (E) Emission spectra and (F) the logarithmic plots of the emission integral of TPM

solid at different excitation intensities (mW) by an 800 nm femtosecond pulsed laser light. CLSM images of HeLa cells incubated with TPM (10 μ M in DMEM). (G) Bright field image; the excitation wavelengths were (H) 405 nm and (I) 800 nm, respectively.

13. Images of HeLa cells at different delay time irradiated with 800 nm laser are required.

Response: We thank the reviewer's suggestion. Images of HeLa cells at different delay time irradiated with 800 nm laser is very necessary. However, no commercial confocal microscope with both fluorescence lifetime imaging (FLIM) and two photon imaging channels are available now. We also inquired the instrument supplier, Acousto-optical Modulators (AOM) system with a femtosecond laser was needed to upgrade the confocal microscope. But the upgradation for our confocal microscope cannot be achieved currently. Therefore, the measurement cannot be achieved now.

Response to Reviewer 3

1. The work is well performed and, except for some sentences (i.e. ...and all do not process potential AIE property..., ... The Stokes shift of TPM..., ... such amorphous nanoparticles can be greatly reduced or even disappeared... just to mention few examples), well and clearly written.

Response: We thank the reviewer's suggestion. We have modified the relevant sentences in the manuscript (...and most of them do not possess potential AIE property..., ...The Stokes shift of TPM..., such amorphous nanoparticles can be greatly weakened).

2. The subject is certainly of large interest; however I have some concerning regarding its novelty. The main interesting aspect of the work is certainly to be associated with the high crystallinity of nanoparticles (allowing their use in bioimaging) which are

usually obtained as “Previous results of organic RTP nanoparticles were amorphous once prepared by the nanoprecipitation methods.^{49,50}”. If this study represents a true novelty on this point, this should be emphasized by indicating that this is the first example or one of the few examples (citing the proper literature).

Response: We thank the reviewer’s suggestion. The preparation of nanoparticles is crucial for bioimaging. This is because the triplet excited energy was sensitive to the oxygen in water. The RTP performance of organic materials was strongly determined by the molecular packing. The nanoparticles with highly crystalline structures and strongly intermolecular interaction showed high phosphorescence efficiency. However, it is still a formidable challenge to obtain RTP nanoparticles with comparable phosphorescence performance with those in crystalline powders. For example, in previous work (*J. Am. Chem. Soc.* 2019, 141, 5045), the phosphorescence lifetime of amorphous nanoparticles was much shorter than that of crystalline powders, and efficiency of amorphous nanoparticles was decreased to one-fourth compared with that of crystalline powder. In order to improve RTP performance, complicated processes were needed (such as ultrasonication or coating with hollow mesoporous silica nanoparticles) for the formation of crystalline structures (*Small* 2020, 16, 1906733; *Angew. Chem. Int. Ed.* 2017, 56, 12160). In our work, uniform size, well-dispersed, highly crystalline nanoparticles were prepared by a simple nanoprecipitation method (Figure S11 and Figure S13). The nanoparticles showed a comparable phosphorescence lifetime and efficiency as crystalline powders due to their high crystallinity. The nanoparticles can be successfully applied for time-resolved bioimaging of HeLa cells and afterglow bioimaging in living nude mice with a high signal-to-noise ratio (Figure 4 and Figure 5).

Figure S11. TEM images of (A) TPM and (B) TPM-Cl nanoparticles in the DMSO/PBS mixtures (10 μ M, DMSO: PBS = 10: 90 volume ratio).

Figure S13. XRD patterns of TPM and TPM-Cl in crystalline powders, after grinding and nanoparticles.

3. The sentence “To our delight, the prepared nanoparticles of TPM and TPM-Cl show comparable phosphorescence quantum yield as crystalline powders due to high crystallinity.” should be supported with proper references. As mentioned, I would substitute “with our delight” with proper examples...

Response: We thank the reviewer’s suggestion. We have cited the proper references in the revised manuscript. Crystallization is an effective strategy to increase the

luminescence efficiency of many AIEgens, as the dense packing of molecules minimizes the intramolecular motions through crystallization.^{44,45} The prepared nanoparticles of TPM and TPM-Cl show comparable phosphorescence quantum yield as crystalline powders due to high crystallinity.

REVIEWERS' COMMENTS

Reviewer #1 (Remarks to the Author):

The resubmitted manuscript has been greatly improved. The authors have exhaustively addressed the issues and made the requested modification. The potential application in vivo has also been added. In my opinion, the paper is now suitable for publication in Nature Communications.

Reviewer #2 (Remarks to the Author):

The authors have adequately revised their manuscript according to my previous comments and suggestions. The quality of the manuscript has been improved after the revision. I do not have further criticism of the work.

Reviewer #3 (Remarks to the Author):

the authors have addressed to all the points raised by the reviewers and, in my opinion, the manuscript can be published in the present form